# Integrative multi-omics analysis of muscle-invasive bladder cancer identifies prognostic biomarkers for frontline chemotherapy and immunotherapy

Qianxing Mo [1✉], Roger Li [2], Dennis O. Adeegbe[3], Guang Peng [4] & Keith Syson Chan [5]

Only a subgroup of patients with muscle-invasive bladder cancer (MIBC) are responders toward cisplatin-based chemotherapy and PD-L1 blockade immunotherapy. There is a clinical need to identify MIBC molecular subtypes and biomarkers for patient stratification toward the therapies. Here, we performed an integrative clustering analysis of 388 MIBC samples with multi-omics data and identified basal and luminal/differentiated integrative subtypes and derived a 42 gene panel for classification of MIBC. Using nine additional gene expression data ($n = 844$), we demonstrated the prognostic value of the 42 basal-luminal genes. The basal subtype was associated with worse overall survival in patients receiving no neoadjuvant chemotherapy (NAC), but better overall survival in patients receiving NAC in two clinical trials. Each of the subtypes could be further divided into chr9 p21.3 normal or loss subgroup. The patients with low expression of *MTAP/CDKN2A/2B* (indicative of chr9 p21.3 loss) had a significantly lower response rate to anti-PD-L1 immunotherapy and worse survival than the patients with high expression of *MTAP/CDKN2A/2B*. This integrative analysis reveals intrinsic MIBC subtypes and biomarkers with prognostic value for the frontline therapies.

[1] Department of Biostatistics & Bioinformatics, H. Lee Moffitt Cancer Center & Research Institute, Tampa, FL 33612, USA. [2] Department of Genitourinary Oncology, H. Lee Moffitt Cancer Center & Research Institute, Tampa, FL 33612, USA. [3] Department of Immunology, H. Lee Moffitt Cancer Center & Research Institute, Tampa, FL 33612, USA. [4] Department of Clinical Cancer Prevention, The University of Texas MD Anderson Cancer Center, Houston, Texas 77030, USA. [5] Department of Pathology and Samuel Oschin Comprehensive Cancer Institute, Cedars-Sinai Medical Center, Los Angeles, CA 90048, USA. ✉email: qianxing.mo@moffitt.org

Bladder cancer is the most common urinary tract malignancy with approximately 549,000 new cases and 200,000 deaths worldwide in 2018[1]. Urothelial cell carcinoma is the predominant histological type, which is often classified as non-muscle invasive bladder cancer (NMIBC), or muscle invasive bladder cancer (MIBC) depending on if tumor has invaded into the muscularis propria. Although only about 25% of newly diagnosed patients present with muscle invasive disease, MIBC has accounted for the majority of bladder cancer mortality[2]. In the past several decades, no major progress was made in the treatment of MIBC and the standard-of-care was limited to chemotherapy and radical cystectomy[3–5]. Recently, immune checkpoint inhibitors were used in clinical trials to treat metastatic bladder cancer, representing a new therapeutic direction[6–10].

MIBC is a heterogeneous disease with poor survival (5-year survival < 50%)[11]. High throughput molecular profiling studies have provided a great opportunity to study the heterogeneity of MIBC. In an effort to identify subgroups of patients that may benefit from systemic therapies, researchers classified MIBC into different molecular subtypes based on various gene expression (GE) signatures. For instance, Volkmer et al. identified a subgroup of MIBC patients who were characterized by high expression of cytokeratins (KRT14, KRT5) and cell surface receptors (THY1, CD44) and were associated with worse overall survival[12]. For this reason, this type of tumors was named as basal subtype[13]. Damrauer et al. and Rebouissou et al. classified MIBC into basal-like and non-basal-like (or luminal) subtypes, while Choi et al. classified MIBC into basal, luminal, and p-53 like subtypes[14–16]. Recently, Mo et al. developed a tumor differentiation classifier that stratified MIBC patients into basal and differentiated subtypes, which were consistent in predicting patient survival in multiple MIBC cohorts[17]. Additionally, other research groups including The Cancer Genome Atlas (TCGA) research network and the bladder cancer molecular taxonomy group further classified MIBC into 4 or more subtypes[18–22]. However, it was generally agreed that MIBC could be basically classified into two major subtypes: basal/squamous-like (BASQ) that shows basal/stem cell features and differentiated/luminal that shows differentiated urothelial cell features[23].

TCGA generated multi-omics data including somatic mutation, GE, DNA copy number, and methylation for over 400 MIBC samples, which were a great resource to study MIBC subtypes[20]. However, these multi-omics data have not been fully integrated to identify MIBC subtypes. In a traditional clustering analysis, a certain number of omics features are selected for platform-specific analysis, which usually include informative and non-informative features. As a result, the clustering outcomes could vary by the selected features and clustering algorithms. To integrate multi-platform clustering results, a second clustering analysis (cluster of cluster analysis) based on the platform-specific results is usually performed[24]. This step-wise clustering approach does not fully take advantage of the inherent structure of multi-omics data and thus may miss the opportunity to reveal the driving factors that determine the tumor subtypes. We hypothesized that a truly integrative clustering (iCluster) analysis of MIBC multi-omics data will be beneficial to discover the inherent driving factors that are essential for tumor classification. The iCluster methods are powerful tools that have been widely used by TCGA and other research groups to characterize a variety of cancers[25–28]. The major characteristics of the iCluster methods are that they can incorporate multi-omics data into a statistical model to perform sample clustering, and at the same time identify driver (informative) omics features from passenger (non-informative) features, which overcome the limits of the platform-specific and step-wise clustering approach[24]. Therefore, in this study, we aimed to use the state-of-the-art iClusterBayes method

to identify MIBC integrative subtypes (iSubtypes) and subtype-specific biomarkers and to evaluate their clinical relevance[28].

## Results

**Integrative molecular subtypes of MIBC.** Integrative clustering analysis of TCGA MIBC samples identified two iSubtypes, which were characterized by distinct molecular patterns across somatic mutation, DNA copy number, methylation and GE (Fig. 1a). We named them as integrative basal (iBasal) and iLuminal (or iDifferentiated) subtypes because they were characterized by the basal and luminal/differentiated GE signature (more details in the following sections)[17]. The top mutated genes that contributed to defining the subtypes included TP53, KDM6A, RB1, FGFR3, TBC1D12, ELF3, and NFE2L2 (Fig. 1a). The iBasal subtype was characterized by higher frequencies of mutations in TP53 (59% vs. 43% in iLuminal, P = 0.0015, Fisher exact test), RB1 (27% vs. 10% in iLuminal, P = 2.53E-5), and NFE2L2 (10% vs. 3% in iLuminal, P = 0.0042). In contrast, the iLuminal subtype was characterized by higher frequencies of mutations in KDM6A (32% vs. 19% in iBasal, P = 0.0053, Fisher exact test), FGFR3 (20% vs. 7% in iBasal, P = 0.00023), TBC1D12 (17% vs. 7% in iBasal, P = 0.0048) and ELF3 (16% vs. 6% in iBasal, P = 0.0041). A region in chr9 p21.3 containing interferon alpha (IFNA) genes, MTAP and CDKN2A/2B was found to be the major contributor for sample clustering, which could be divided into copy number normal (9p21.3 N) and loss (9p21.3 L) regions (Fig. 1a copy number). Compared to the 9p31.3 L group, the 9p21.3 N group was associated with higher frequencies of mutations in TP53 (60% vs. 39% in 9p21.3 L, P = 6.95E-5, Fisher exact test) and RB1 (29% vs. 5% in in 9p21.3 L, P = 1.71E-10), but a lower frequencies of mutations in FGFR3 (8% vs. 21% in 9p21.3 L, P = 6.98E-4) and NFE2L2 ( < 1% vs. 12% in 9p21.3 L, P = 6.93E-7). However, overall, the iSubtypes and their 9p21.3 L/N subgroups were not significantly associated with tumor mutation burden (Supplementary Fig. 1). The subtype-driver genes in the methylation data formed two clusters M1 and M2 with opposite methylation patterns (Fig. 1a methylation). Gene ontology (GO) term enrichment analysis showed that methylation cluster M2 was most enriched with genes involved in cell adhesion/motion/morphogenesis, response to wounding/organic substance, leukocyte activation, and skeletal/urogenital system development, while no significant biological process was found in cluster M1 (Summarized in Fig. 1b and Supplementary data 1). The subtype-driver genes in the mRNA data also formed 2 major clusters E1 and E2 (Fig. 1a mRNA expression). The GE pattern in E1 and methylation pattern in M2 appeared to be negatively correlated. The major enriched GO terms in M2 were also found to be enriched in E1, while E2 was most enriched with genes involved in metabolic processes (Summarized in Fig. 1c and Supplementary data 1).

**Comparison of MIBC iSubtypes with GE-based subtypes.** Since most molecular subtypes were defined by GE, likely with a more direct effect on biological functions, we compared the iSubtypes with the previously reported GE subtypes. TCGA classified the MIBC samples into five subtypes including basal-squamous (35%), luminal-papillary (35%), luminal-infiltrated (19%), luminal (6%), and neuronal (5%)[20]. Recently, Kamoun et al. derived 6 consensus GE subtypes using 6 published classifiers, namely basal/squamous (Ba/Sq), luminal papillary (LumP), luminal nonspecified (LumNS), luminal unstable (LumU), stroma-rich, and neuroendocrine-like (NE-like). These subtypes were generally concordant with the two major basal and luminal subtypes reported by TCGA and other groups[14–16,21]. Mo et al. also reported two GE subtypes, namely basal and differentiated[17]. Interestingly, we found that iSubtypes and the tumor differentiation subtypes were highly concordant, with an overall concordance rate of 86% (Fig. 2a). Compared to the TCGA and

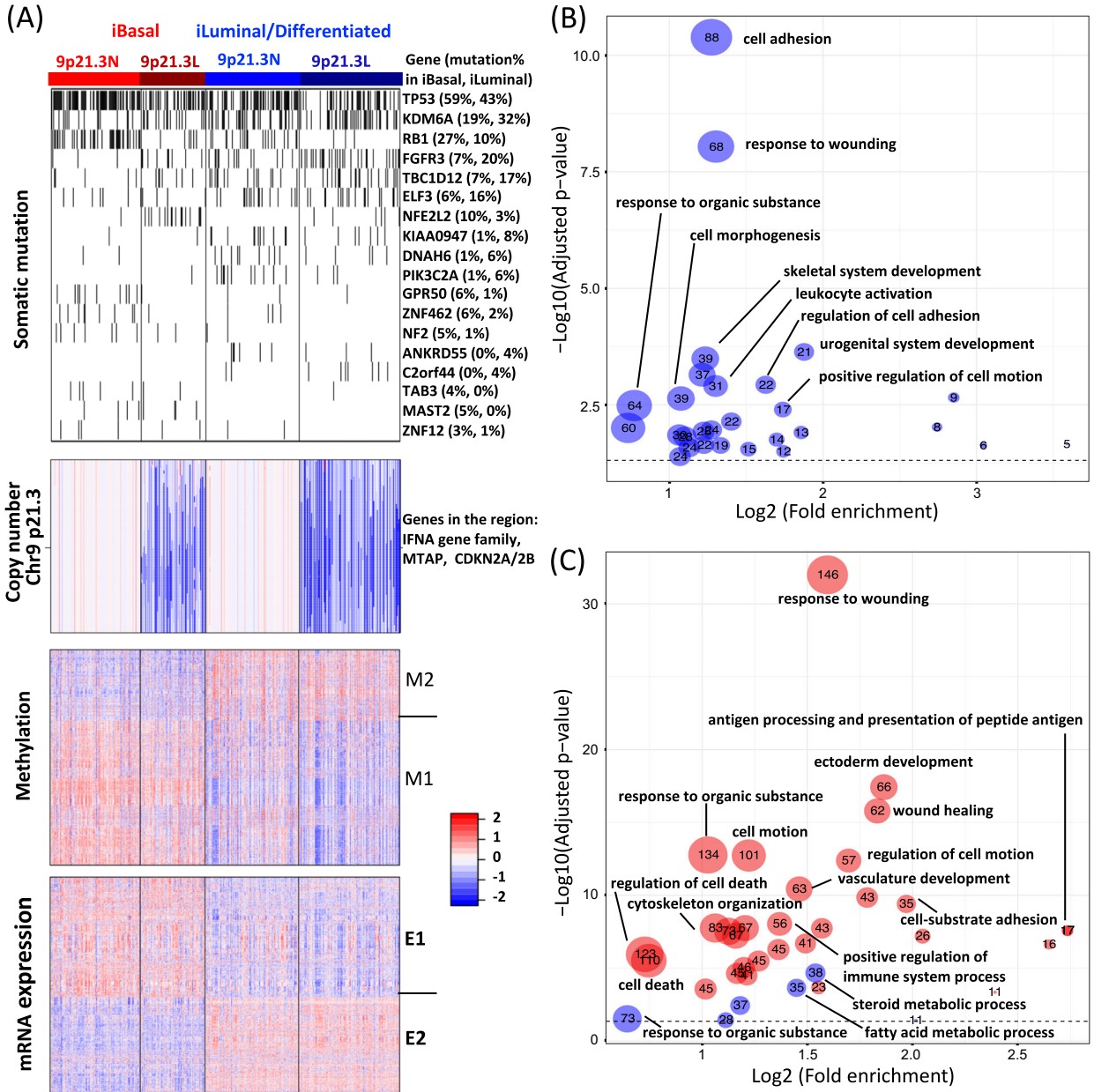

**Fig. 1 Integrative subtypes of MIBC. a** Heatmaps of driver genomic features. **Somatic mutation**: mutated and normal genes are represented by black and white colors, respectively. **Copy number**: red, white, and blue represent potential copy number gain, normal, and loss, respectively. **Methylation**: red and blue represent hypermethylation and hypomethylation, respectively; Driver genes were grouped into two clusters M1 and M2. **mRNA**: red and blue represent high and low expression, respectively. Driver genes were grouped into two clusters E1 and E2. **b** Top enriched biological processes in methylation cluster M2. **c** Top enriched biological processes in mRNA expression clusters E1 (red dots) and E2 (blue dots). The numbers of genes belonging to biological processes are shown in the dots.

consensus subtypes, the iBasal subtype highly overlapped with the basal-squamous and neuronal (or neuroendocrine-like) subtypes, while the iLuminal subtype highly overlapped with the TCGA luminal, luminal-papillary and luminal-infiltrated subtypes and the consensus LumP, LumU, LumNS, and Stroma-rich subtypes (Fig. 2a). The results show a high consistency as well as some variability among subtype assignments by different signatures. In the patients receiving no NAC, the iBasal subtype was significantly associated with worse overall survival, compared to the iLuminal subtype (Fig. 2b, $P = 0.00042$, Log-rank test). Their survival curves (median survival, iBasal: 22.5 vs. iLuminal: 54.9 months; Fig. 2b) were more widely separated than the curves of the tumor differentiation subtypes[17] (median survival, basal: 25.6 vs. differentiated:

41.7 months; Fig. 2c, $P = 0.015$, Log-rank test), demonstrating an improvement of patient classification in terms of survival. The iSubtypes were significantly associated with pathologic stage ($P = 0.011$, Fisher exact test), T stage ($P = 0.0017$), and gender ($P = 0.027$) (Supplementary Table 1). However, multivariate Cox regression analysis showed that the iSubtype was an independent predictor of overall survival when the baseline variables including age, gender and smoking status, pathologic stage (or T stage) were included into the models (Supplementary Table 2). Although there were 5 TCGA subtypes, only the luminal-papillary survival curve was significantly separated from the others (Fig. 2d, overall $P = 0.00047$, Log-rank test). There was no significant difference in overall survival among the basal-squamous, luminal, luminal-infiltrated, and neuronal

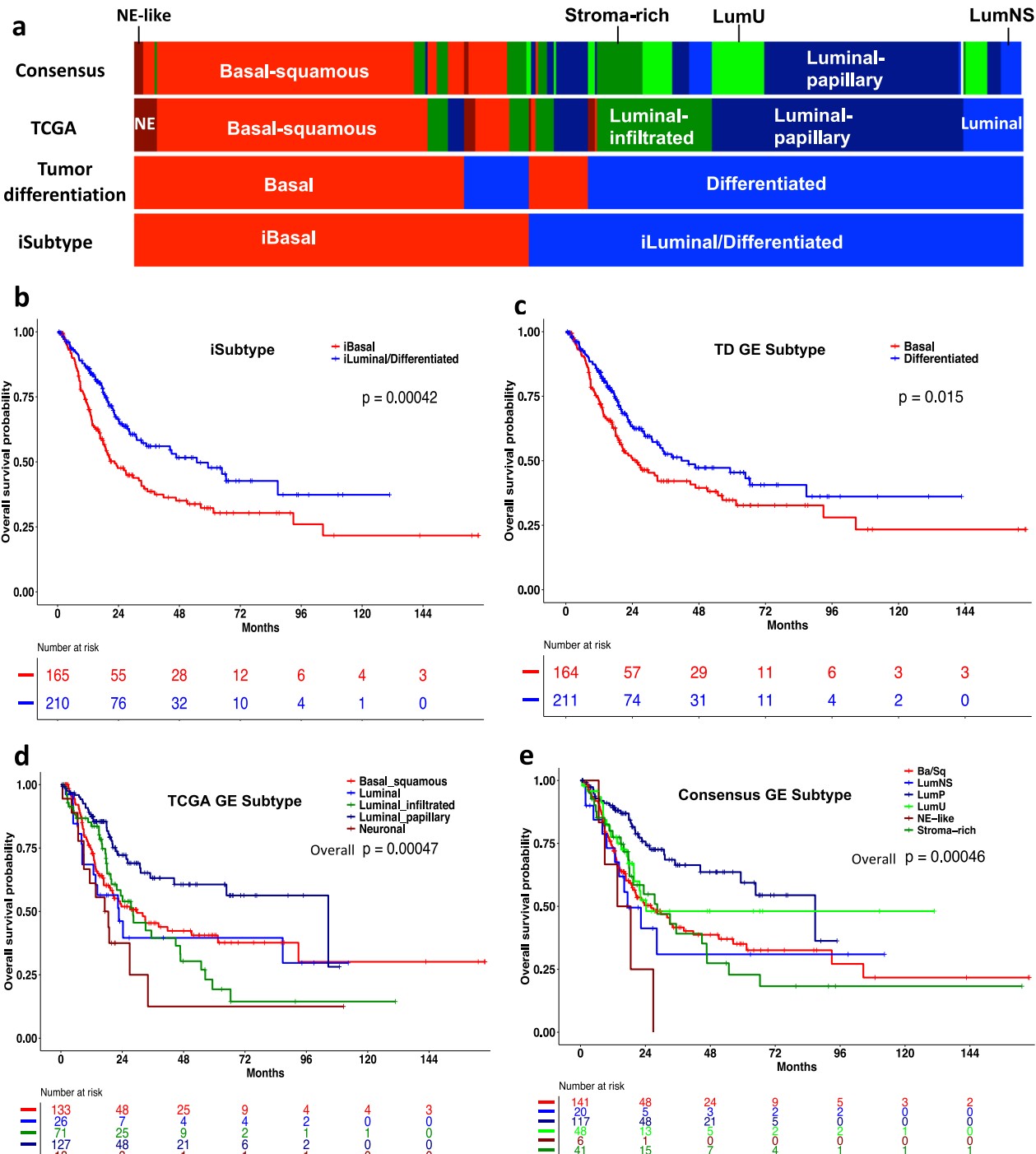

**Fig. 2 Comparison of iSubtypes with other GE based subtypes in the TCGA cohort. a** TCGA samples ($n = 388$) colored according to the iSubtypes, TD, TCGA, and consensus GE subtypes, respectively. **b** Patient overall survival stratified by iBasal and iLuminal/Differentiated subtypes. **c** Patient overall survival stratified by the basal and differentiated subtypes defined by the TD GE signature. **d** Patient overall survival stratified by the TCGA GE subtypes. Luminal_papillary vs. the others: $p < 0.0001$. NE = neuronal. **e** Patient overall survival stratified by the consensus GE subtypes. LumP vs. the others: $p <$ 0.0001. Ba/Sq: basal/squamous; LumNS= luminal nonspecified; LumP= luminal papillary; LumU= luminal unstable; NE = neuroendocrine. Kaplan–Meier curves and log-rank test $p$-values shown on **b**–**e** are based on 375 non-NAC patients without missing data.

subtypes (Fig. 2d, $P = 0.3$, Log-rank test). Similarly, although there were 6 consensus subtypes, only LumP was significantly different from the other 5 subtypes (Fig. 2e, overall $P = 0.00046$, Log-rank test). There was no significant difference in overall survival among the Basal-squamous, LumNS, LumU, NE-like, and Stroma-rich subtypes (Fig. 2e, $P = 0.3$, Log-rank test). There were 10 patients with a history of NAC in this cohort of patients in which 6 patients were

clustered to the iBasal subtype and 4 patients were clustered to the iLuminal subtype (Supplementary Table 2). Similar results were obtained when they were included in the survival analyses (Supplementary Fig. 2).

**Subtype-specific survival and responses to PD-L1 blockade immunotherapy.** The iCluster analysis identified 42 classical

basal-luminal markers as the subtype drivers, which included the 18 tumor differentiation (TD) genes (Fig. 3a)[17]. In general, the basal markers had a higher expression in the basal subtypes, while the luminal markers had a higher expression in the luminal subtype (Fig. 3a, left panel). To investigate whether these markers had similar expression patterns in other MIBC cohorts, we analyzed another RNA-seq data from a large phase 2 trial (named IMvigor210, $n = 348$) that studied the clinical activity of PD-L1 blockade with atezolizumab in patients with locally advanced and metastatic urothelial cancer[6]. The RNA-seq data were generated from pre-treatment tumor samples of the patients. Using the TCGA data as the training set and the IMvigor210 data as the testing set, we classified the IMvigor210 samples into 2 subtypes using the k-nearest neighbor (KNN) method and the 18 TD genes and the 42 basal-luminal genes, respectively. The classification results of the IMvigor210 samples based on the 42 genes are shown in Fig. 3a (right panel). The resulting subtypes had very similar expression pattern as the TCGA samples (Fig. 3a, left panel). Furthermore, the basal subtype defined by the 42 basal-luminal genes was associated with worse survival, compared to the luminal subtype (Fig. 3b, $P = 0.041$, Log-rank test). Remarkably, the 18 TD genes had similar power in classifying the patient samples into 2 subtypes with distinct survival (Fig. 3c, $P = 0.04$, Log-rank test).

As described previously, the basal and luminal iSubtypes could be further divided into copy number normal (9p21.3 N) and loss (9p21.3 L) regions that contain MTAP and CDKN2A/2B (Fig. 1a copy number). We found that the expression levels of MTAP and CDKN2A/2B were positively correlated with the copy number data of the region in chr9 p21.3 (Fig. 3d TCGA samples). Compared to the 9p21.3 N group, the 9p21.3 L was associated with significant reduction of expression of CDKN2A by 27.35-fold, CDKN2B by 6.83-fold, and MTAP by 6.28-fold (Supplementary Fig. 3). Since the copy number data for the IMvigor210 cohort were not available, we could not further classify the patient samples according to the chr9 p21.3 status. Therefore, we used the 3 genes' expression as a surrogate of chr9 p21.3 status to further classify the IMvigor210 samples into MTAP/CDKN2A/2B high expression (G3High) and low expression (G3Low) groups (Fig. 3d IMvigor210 samples). Compared to the G3High group, the G3Low group was associated with a significant reduction of expression of CDKN2A by 29.08-fold, CDKN2B by 6.27-fold, and MTAP by 4.22-fold, which were quite consistent with the results observed in the TCGA data (Supplementary Fig. 3). There was no significant difference in overall survival between the G3Low and G3High groups in the TCGA cohort (Fig. 3e, $P = 0.35$, Log-rank test) and in the basal (Fig. 3f, $P = 0.2$) and luminal sub-cohorts (Fig. 3f, $P = 0.6$), while the overall difference in overall survival was primarily driven by the basal and luminal subtypes themselves (Fig. 3f, $P = 0.0021$).

The patients in the IMvigor210 cohort underwent PD-L1 blockade therapy. We found that the complete or partial (CR/PR) response rate of the PD-L1 blockade therapy of the G3High groups were at least 2-fold higher than the response rate of the G3Low group in the IMvigor210 cohort (Fig. 4a, $P = 0.00084$, Fisher exact test). The G3High group was also associated with a higher expression of PD-L1 (Fig. 4b, $P = 0.011$, t-test), and better overall survival (Fig. 4c, $P = 0.0043$, Log-rank test), compared to the G3Low group. Similar results were also observed in the basal and luminal sub-cohorts, respectively (Fig. 4d–f). Mariathasan et al. reported that the immune cell (IC) subtypes, the tumor-immune phenotypes, and the Lund subtypes were associated with responses to PD-L1 blockade therapy[29]. For a head-to-head comparison, we showed their results in Fig. 4. In general, for the immune cell and tumor-immune subtypes, a higher PD-L1

expression level was associated with a higher response rate, and better overall survival (Fig. 4g–l), which were consistent with our findings. Interestingly, the strict positive correlation between the response rate and PD-L1 expression was not observed in the Lund subtypes of Sjodahl et al.[19], although the genomically unstable (GU) subtype with the highest response rate was associated with the best overall survival (Fig. 4m–o). Overall, our results were comparable to the results reported by Mariathasan et al.[29].

**Prognostic value of subtype-specific GE signature in multiple cohorts of study.** To further test the prognostic power of the basal and luminal signature revealed in the TCGA data, we collected 8 additional microarray GE data sets. Six microarray data sets were from retrospective studies in which the detailed information for chemotherapy was not publicly available[14,16,19,30–32]. Two microarray data sets were from clinical trials that were designed to evaluate the effect of NAC (MVAC or MVAC + B) on clinical outcomes and the impact of MIBC subtypes[16,31]. Using the TCGA GE data as the training data set, we classified the samples in the other 8 data sets into two subtypes based on the TD (18 genes) and basal-luminal (42 genes) signature, respectively. Figure 5a, b shows the classification results based on 42 basal-luminal genes. It can be seen that the GE patterns observed in the TCGA cohort were also observed in the 8 independent cohorts. In addition, the basal subtype defined by the 42 basal-luminal GE signature was associated with worse overall survival, compared to the luminal subtype in the 6 retrospective cohorts (Fig. 5c, $P = 0.013$, Log-rank test), which was similar to the results observed in the TCGA and IMvigor210 cohorts and consistent with other published results[12,14–16]. In contrast, the basal subtype defined by 42 basal-luminal GE signature was associated with better overall survival (Fig. 5d, $P = 0.017$, Log-rank test) in the two clinical trial cohorts, implying basal subtype responded better to the NAC than the luminal subtype[31,33]. Similarly, the TD GE signature also had excellent discriminating power in classifying MIBC samples into subtypes with distinct survival in the 6 retrospective cohorts (Fig. 5e, $P = 0.017$, Log-rank test) and the 2 clinical trial cohorts (Fig. 5f, $P = 0.081$, Log-rank test).

**Pathways and gene sets enriched in the basal and luminal subtypes.** In an effort to identify the pathways that were altered between the basal and luminal subtypes, we performed GSEA on the well-established BIOCARTA and KEGG pathways using the TCGA data. Figure 6 shows the GE patterns of the top 20 up-regulated BIOCARTA pathways in the basal and luminal subtypes. Strikingly, 16 of the 20 up-regulated pathways in the basal subtype were related to immune network, 2 pathways were related to cell cycle regulation, and 2 pathways were related to cell motility. Consistently, most of the top up-regulated KEGG pathways in the basal subtype were related to immune network, and pathways related to cell adhesion and signal transduction were also among the top list (Supplementary Fig. 4). Interestingly, the top up-regulated KEGG pathways in the luminal subtype were primarily related to various metabolisms (Supplementary Fig. 5).

To gain further insights about the immune network between the basal and luminal subtypes, we investigated if the subtypes were differentially enriched with tumor-infiltrating lymphocytes (TILs) by performing GSEA on immune-cell-specific gene sets[34]. The GSEA results showed that the innate immune cells including macrophages (FDR < 0.0001, gene-based permutation test) and neutrophils (FDR = 0.0014) and adaptive immune cells including T cells (FDR = 0.014), T helper 1 (Th1) cells (FDR < 0.0001), Th2 cells (FDR = 0.05) and Cytotoxic cells (FDR = 0.05) were

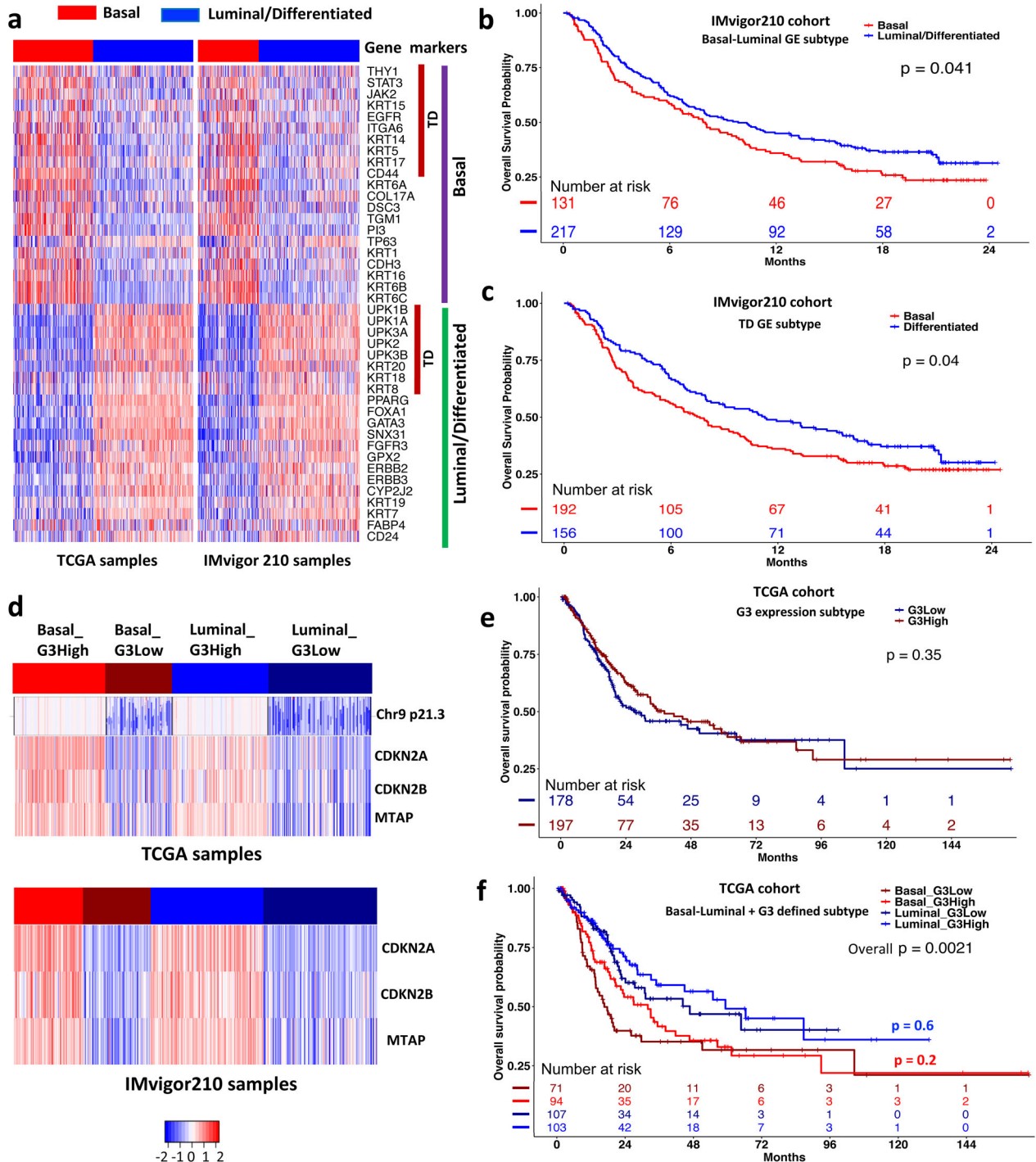

**Fig. 3 Prognostic power of tumor differentiation (TD) and basal-luminal gene expression signatures. a** Heatmaps of gene expression of 42 basal-luminal genes (including 18 TD genes) in the TCGA ($n = 388$) and IMvigor210 ($n = 348$) cohorts. **b**, **c** Patient overall survival stratified by the basal and luminal subtypes defined by the 42 basal-luminal genes (**b**), and by the basal and differentiated subtypes defined by the 18 TD genes (**c**) in the IMvigor210 cohort. **d** Heatmaps of G3 (*MTAP/CDKN2A/2B*) expression in the basal and luminal/differentiated subtypes in the TCGA and IMvigor210 cohorts, which were further divided into relatively G3 high expression (G3High) or low expression (G3Low) subgroup. On the top panel, the copy number data of chr9 p21.3 are aligned with the G3 expression data in the TCGA cohort, indicating a high concordance between copy number and gene expression. **e** Overall survival stratified by the G3High and G3Low groups in the TCGA cohort (375 non-NAC patients). **f** Overall survival stratified by the Basal_G3High, Basal_G3Low, Luminal_G3High, and Lumina_G3Low groups in the TCGA cohort (375 non-NAC patients). Log-rank test was used to compare subtype-specific survival curves.

significantly enriched in the basal subtypes (Fig. 7a), while dendritic cells were not (FDR = 0.41). This analysis crudely implied that the basal tumors were infiltrated with a large spectrum of immune cells to a higher degree than the luminal MIBC. However, it remained puzzling how to rationalize that

both positive and negative regulators of effector T cells were upregulated in the basal MIBC.

**Digital dissection of tumor microenvironment of MIBC.** Next, we digitally dissected the tumor microenvironment by performing

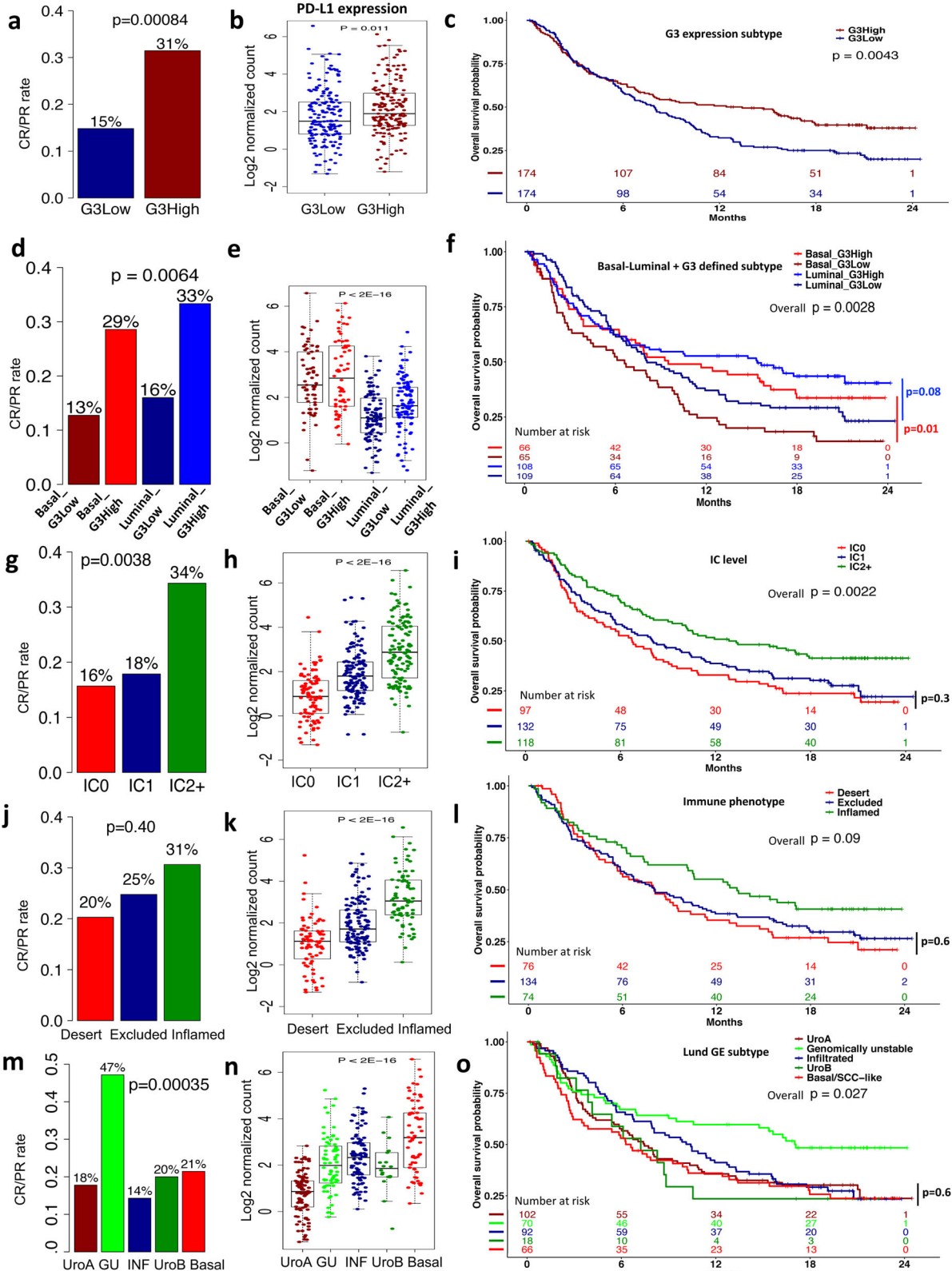

**Fig. 4 Subtype-specific response (CR/PR) rates of PD-L1 blockade therapy, PD-L1 expression levels, and survival in the IMvigor210 cohort. a** Response rates, **b** PD-L1 expression levels, and **c** KM survival curves of the G3High and G3Low groups. **d** Response rates, **e** PD-L1 expression levels, and **f** KM survival curves of the G3High and G3Low groups in the basal and luminal sub-cohorts. **g** Response rates, **h** PD-L1 expression levels, and **i** KM survival curves of the immune cell (IC) subgroups. **j** Response rates, **k** PD-L1 expression levels, and **l** KM survival curves of the tumor-immune phenotypes. **m** Response rates, **n** PD-L1 expression levels, and **o** KM survival curves of the Lund subtypes. Fisher exact test was used to compare the CR/PR frequencies; t-test or ANOVA was used to compare PD-L1 expression between the subtypes. Log-rank test was used to compare subtype-specific survival curves.

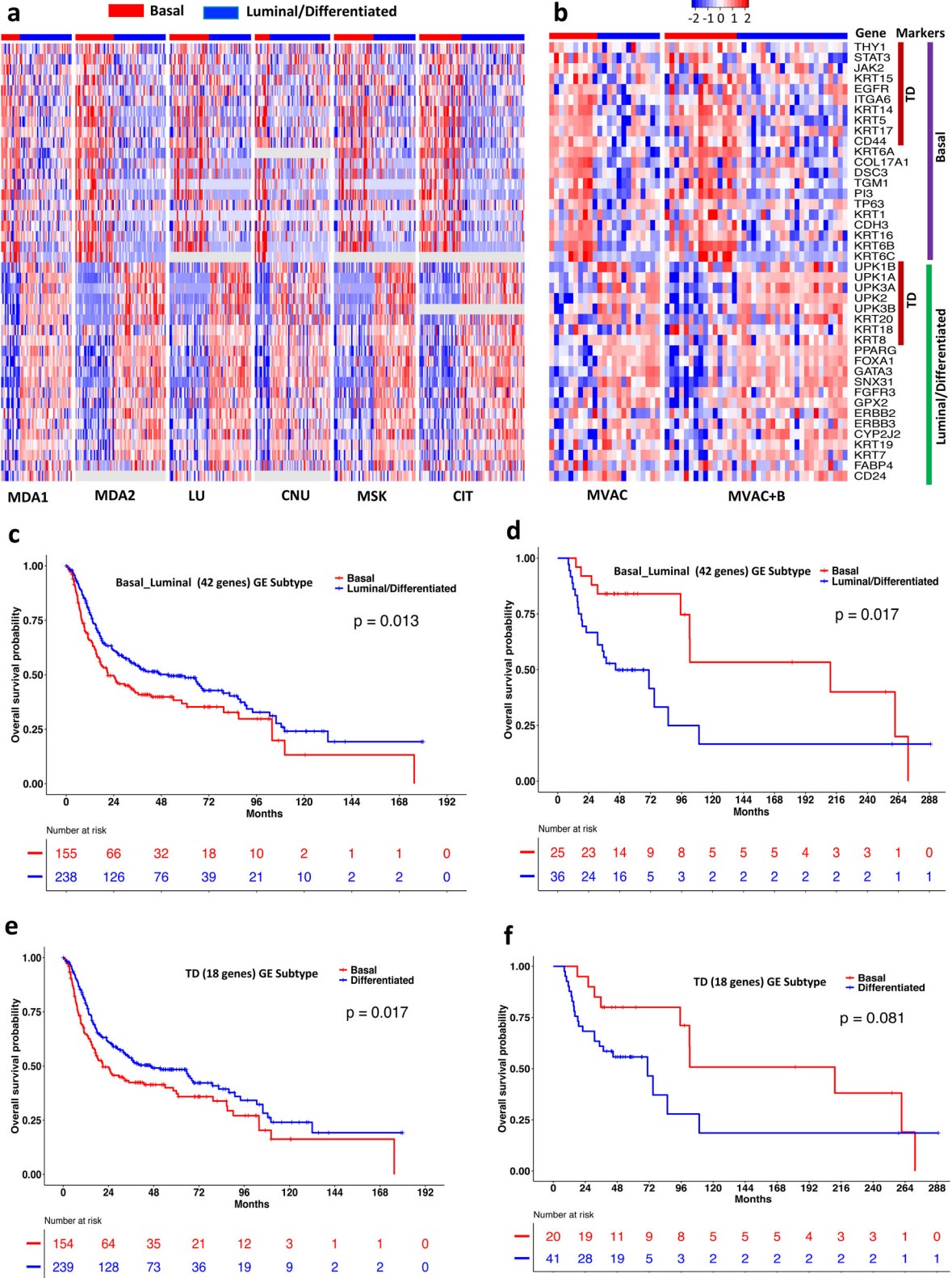

**Fig. 5 Prognostic power of tumor differentiation (TD) and basal-luminal gene expression signatures. a, b** Heatmaps of gene expression of 42 basal-luminal genes (including 18 TD genes) in the 6 retrospective study cohorts (MDA1, MDA2, LU, CNU, MSK, CIT) (**a**) and the 2 clinical trial cohorts (MVAC, MVAC + B) (**b**). **c, d** Patient overall survival stratified by the basal and Luminal subtypes defined by the 42 basal-luminal genes in the 6 retrospective study cohorts (**c**) and the 2 clinical trial cohorts (**d**). **e, f** Patient overall survival stratified by the basal and differentiated subtypes defined by the 18 TD genes in the 6 retrospective study cohorts (**e**) and the 2 clinical trial cohorts (**f**). MDA1: MD Anderson Cancer Center (USA) cohort 1; MDA2: MDA cohort 2; LU: Lund University, Sweden; CNU: Chungbuk National University, South Korea; MSK: Memorial Sloan-Kettering Cancer Center, USA; CIT: Curie Institute, France; MVAC: methotrexate, vinblastine, doxorubicin, and cisplatin; B: bevacizumab. Log-rank test was used to compare subtype-specific survival curves.

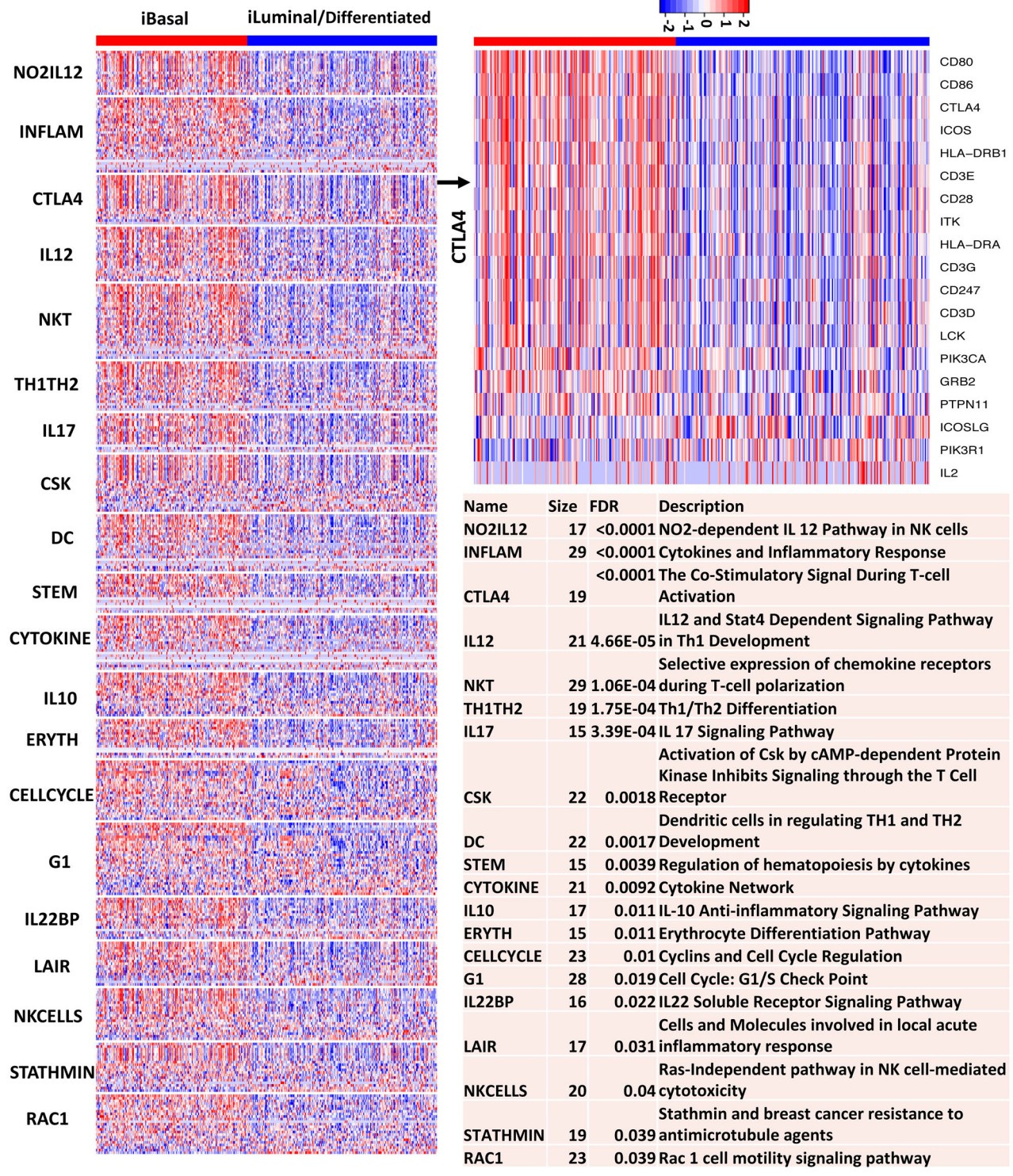

**Fig. 6 Top up-regulated BIOCARTA pathways in the basal subtype of the TCGA MIBC samples.** On the heatmap, red and blue represent high and low expression, respectively. Gene-based permutation test was used to calculate the FDR.

| Name | Size | FDR | Description |
|---|---|---|---|
| NO2IL12 | 17 | <0.0001 | NO2-dependent IL 12 Pathway in NK cells |
| INFLAM | 29 | <0.0001 | Cytokines and Inflammatory Response |
| CTLA4 | 19 | <0.0001 | The Co-Stimulatory Signal During T-cell Activation |
| IL12 | 21 | 4.66E-05 | IL12 and Stat4 Dependent Signaling Pathway in Th1 Development |
| NKT | 29 | 1.06E-04 | Selective expression of chemokine receptors during T-cell polarization |
| TH1TH2 | 19 | 1.75E-04 | Th1/Th2 Differentiation |
| IL17 | 15 | 3.39E-04 | IL 17 Signaling Pathway |
| CSK | 22 | 0.0018 | Activation of Csk by cAMP-dependent Protein Kinase Inhibits Signaling through the T Cell Receptor |
| DC | 22 | 0.0017 | Dendritic cells in regulating TH1 and TH2 Development |
| STEM | 15 | 0.0039 | Regulation of hematopoiesis by cytokines |
| CYTOKINE | 21 | 0.0092 | Cytokine Network |
| IL10 | 17 | 0.011 | IL-10 Anti-inflammatory Signaling Pathway |
| ERYTH | 15 | 0.011 | Erythrocyte Differentiation Pathway |
| CELLCYCLE | 23 | 0.01 | Cyclins and Cell Cycle Regulation |
| G1 | 28 | 0.019 | Cell Cycle: G1/S Check Point |
| IL22BP | 16 | 0.022 | IL22 Soluble Receptor Signaling Pathway |
| LAIR | 17 | 0.031 | Cells and Molecules involved in local acute inflammatory response |
| NKCELLS | 20 | 0.04 | Ras-Independent pathway in NK cell-mediated cytotoxicity |
| STATHMIN | 19 | 0.039 | Stathmin and breast cancer resistance to antimicrotubule agents |
| RAC1 | 23 | 0.039 | Rac 1 cell motility signaling pathway |

xCell analysis of 64 cell types for each TCGA MIBC sample[35]. The xCell score is based on single-sample GSEA (ssGSEA) and can be used to measure the enrichment of a set of genes among the top of a ranked GE profile. Overall, the basal subtype had higher immune score (P = 7.89E-12, Wilcoxon rank-sum test) and microenvironment score (P = 3.12E-7) than the luminal subtype (Fig. 7b; Supplementary Fig. 6). As expected, epithelial cells, keratinocytes, and smooth muscle cells were among the top enriched cells in the MIBC samples (Fig. 7b). Epithelial cells and keratinocytes appeared to be more enriched in the basal subtype, consistent with the characteristics of basal cells. The top enriched cells also included immune cells such as actived dendritic cells (aDC), immature DC (iDC), conventional DC (cDC), Th1/Th2 cells, and macrophages. Interestingly, xCell analysis revealed enrichment of dendritic cell content in basal MIBCs,

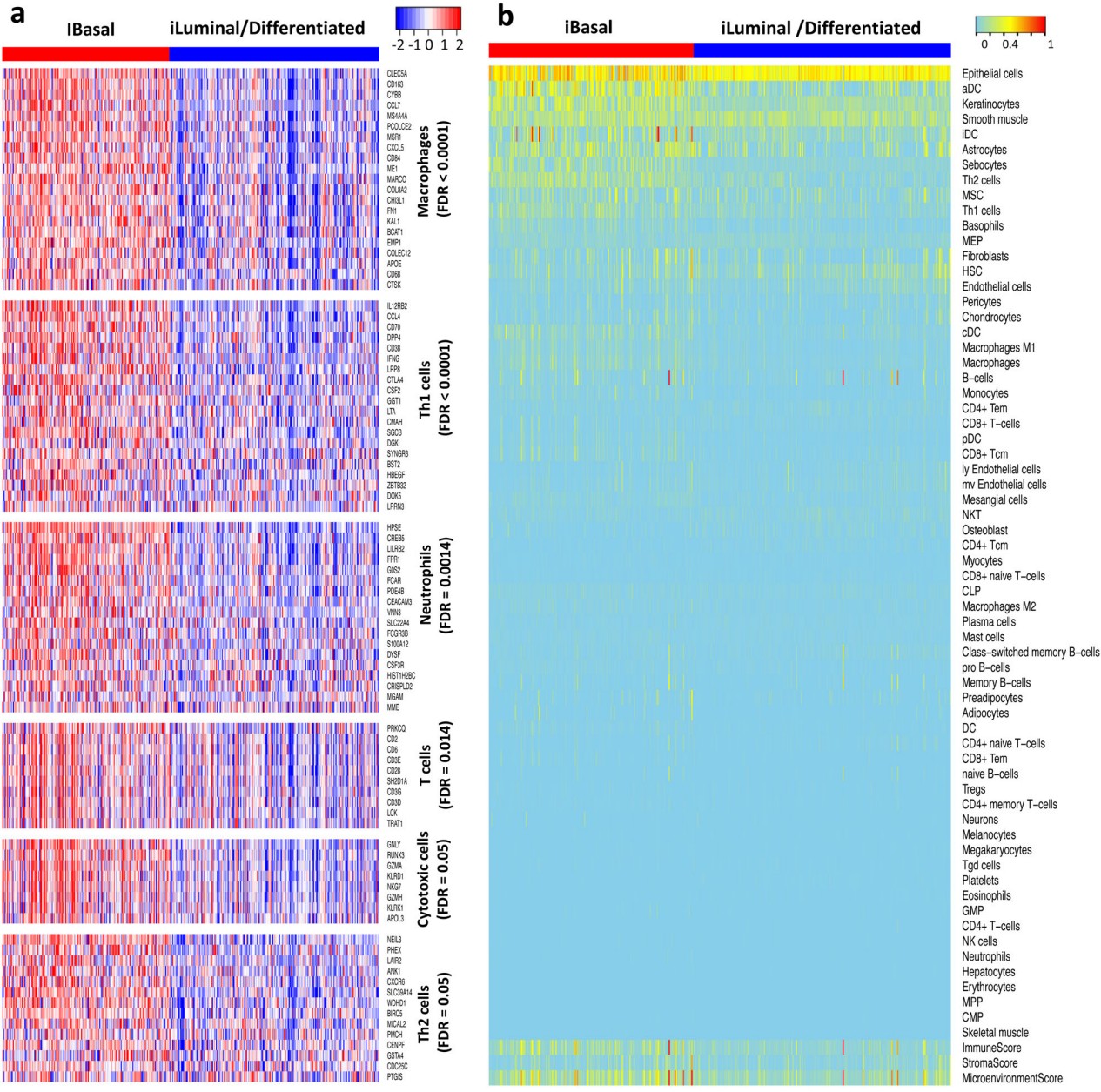

**Fig. 7 Digital dissection of TCGA MIBC samples. a** Immune-cell-specific gene expression. On the heatmaps, red: high expression, blue: low expression. **b** xCell scores of the 64 cell types. On the heatmaps, cyan to red represent low to high xCell scores. Columns represent samples, and rows represent cell types and immune, stroma, and microenvironment scores. Gene-based permutation test was used to calculate the FDR.

while GSEA did not. Another interesting finding was that CD8 + T cells were not heavily infiltrated in basal nor luminal MIBCs (Fig. 7b).

## Discussion

With the aim to identify MIBC subtypes and biomarkers that are clinically relevant, we performed iCluster analysis of MIBC multi-omics data and identified two iSubtypes with chr9 p21.3 loss or normal subgroups that shed light on patient stratification for frontline immunotherapy and chemotherapy. Relatively high mutation rates of *TP53*, *RB1*, and *NFE2L2* were observed in the basal subtype. *TP53* and *RB1* mutations tended to be positively correlated, and were highly mutated in MIBC or high-grade NMIBC[11,36–38]. Therefore, concurrent mutations of *TP53* and *RB1* might contribute to cancer aggressiveness. Cigarette smoking

is a well-known risk factor for bladder cancer. Mutations in *NFE2L2* could limit its effect on the genes that inhibit the damaging effects caused by carcinogens in cigarette smoke[39,40]. In contrast, relatively high mutation rates of *KDM6A* and *FGFR3* were observed in the luminal subtype. Interestingly, *KDM6A* and *FGFR3* were seen at higher rates in NMIBC, compared to MIBC[38]. *KDM6A* encodes a histone demethylase, and thus mutations could lead to gene silencing. Activating *FGFR3* mutations were frequently observed in bladder cancer, leading to morphological transformation and cell proliferation[41,42]. The basal subtype was characterized by high expression of the basal-squamous markers including *STAT3* and *EGFR*, which could be potential therapeutic targets for basal MIBC. In a mouse model, the constitutive expression of *STAT3* led to the development of invasive bladder[43]. Preclinical studies suggested that treatment

efficacy of anti-EGFR could be improved by basal subtype-specific anti-EGFR therapy[14], and phase II clinical trials showed that the EGFR inhibitors had antitumor activity[44,45]. The luminal subtype was characterized by high expression of the differentiated/luminal markers including *FGFR3*, a potential therapeutic target. Early phase clinical trials have shown benefits of anti-FGFR agents in *FGFR3*-mutant bladder urothelial cancer[46–48].

Both the basal and luminal subtypes could be further divided into chr9 p21.3 N (normal) and chr9 p21.3 L (loss) subgroups (Fig. 1a). Chr9 p21.3 region contains genes including *MTAP* and *CDKN2A/2B*. *MTAP* encodes methylthioadenosine phosphorylase, an enzyme essential for the salvage of adenine and methionine in polyamine metabolism. MTAP converts methylthioadenosine to adenine and methylthioribose-1-phosphate, which are used for AMP and methionine synthesis, respectively[49]. Therefore, loss of MTAP can make the generation of AMP dependent on the *de novo* synthesis pathway. As a result, tumor cells lack of MTAP could be potentially sensitive to inhibitors of the de novo purine synthesis pathway[50]. *CDKN2A* and *CDKN2B* encode two important cell cycle regulatory proteins p16 and p15 respectively, which function as inhibitors of cyclin-dependent kinases CDK4 and CDK6, leading to blocking the phosphorylation of the *RB* protein. In an alternative reading frame (ARF), *CDKN2A* encodes protien p14ARF, which can bind to and sequester MDM2 (a E3 ubiquitin ligase), leading to blocking MDM2-mediated degradation of p53[51]. Therefore, *CDKN2A/2B* encodes tumor suppressors capable of inducing cell cycle arrest in G1 and G2 phases. The chr9 p21.3 deletion has been reported in bladder cancer cell lines and tissue samples as well as other cancers including glioma, leukemias, lung cancer, and mesotheliomas[20,49,52–54].

By analyzing the IMvigor210 RNA-seq data, we found that the *MTAP/CDKN2A/2B* low expression (G3Low, indicative of chr9 p21.3 L) groups had significantly lower anti-PD-L1 response rates and worse survival than the G3High (indicative of chr9 p21.3 N) groups. Interestingly, loss of chr9 p21.3 or low expression of the genes in the region was associated with worse survival in patients with NMIBC, melanoma, mesotheliomas, glioma, and oral squamous cell carcinoma[53,55–58], which was consistent with our findings in patients with MIBC. Although PD-L1 had a higher expression in the basal subtype than that in the luminal subtype (Fig. 4e and Supplementary Fig. 7), there was no significant correlation between the 2 major subtypes and the response rates of the anti-PD-L1 immunotherapy in the IMvigor210 cohort (CR/PR rate, basal: 21% vs. luminal: 24%, $P = 0.57$, Fisher exact test). In the Lund subtypes, the basal with the highest median PD-L1 expression did not have the highest anti-PD-L1 response rate (Fig. 4m, n). It was reported that PD-L1 expression on immune cells (measured by the percentages of PDL1-positive immune cells) was significantly correlated with the response[6,7]. However, PD-L1 expression on tumor cells was not correlated with the response[29]. Therefore, an overall measurement of PD-L1 expression in tumor tissues that are often mixed with immune cells may not be informative with respect to the response to immunotherapy. Interestingly, when the basal and luminal subtypes were further divided into G3High/Low groups, we observed the positive correlations among the overall PD-L1 expression levels, anti-PD-L1 response rates and overall survival, consistent with the observations in immune cell subtypes and immune phenotypes reported by Mariathasan et al.[29] (Fig. 4d–l).

The clustering of the TCGA samples to the two major basal and luminal subtypes were primarily driven by the GE and methylation patterns, which were negatively correlated. Currently, GE data are the most popular omics data type and most of the bladder cancer subtypes are defined by GE signatures. In order to make our findings useful in practice, we derived a 42-gene panel including the 18 TD genes as a surrogate of the identified 2655 driver genes (Fig. 1a and Supplementary Fig. 8) for classification and prognostic analysis. The 42 genes consist of the classical basal-luminal genes and are of biological implications. Using the 42 basal-luminal GE and the 18 TD GE signatures, MIBC samples can be classified into basal and luminal/differentiated subtypes. The basal subtype was characterized by relatively high expression of basal-squamous markers (e.g., *CD44*, *KRT5*, *KRT6B/C*, *KRT14*, *TGM1*, *DSC3*, *PI3*), while the luminal subtype was characterized by relatively high expression of luminal markers of terminally differentiated urothelial umbrella cells (*UPK1A/B*, *UPK2*, *UPK3A/B*, *KRT20*, *SNX3*) (Figs. 3a, 5a, b). Using the IMvigor210 RNA-seq data and 8 microarray data sets, we demonstrated that the 42 base-luminal genes and the 18 TD genes had almost equivalent prognostic power in classifying MIBC samples into clinically relevant subgroups (Figs. 3b, c, 5c–f). Interestingly, the basal subtype was associated with worse overall survival in the IMvigor210 and the 6 retrospective study cohorts, which was consistent with previous observations[12,14–17,19]. One may speculate that despite the basal subtype exhibiting signatures suggestive of increased immune cell infiltrates, the balance between inhibitory and stimulatory/effector cellular and molecular mechanisms may be skewed towards the former, a pattern that will contribute to muting potential anti-tumor response and is consistent with the poor survival outcome in patients with this subtype. Comprehensive profiling of the tumor immune microenvironment is, therefore, necessary to better understand the interplay between the tumor-associated immune cells between the basal and luminal subtypes and how this impinges upon potential anti-tumor immunity. In the 2 clinical trials that aimed to evaluate the effects of cisplatin-based NAC on MIBC patients, the basal subtype was associated with better overall survival, consistent with the observations by McConkey et al. and Seiler et al.[31,33]. Our pathway analyses revealed that pathways involved in cell motion/adhesion and cell cycle were up-regulated in the basal subtype, which might reveal the mechanism related to its aggressive nature. Chemotherapy might be able to restrain tumor cell proliferation and motility, leading to the susceptibility of the basal subtype to chemotherapy. In addition, chemotherapy may affect the dynamics of tumor-associated immune cells either in a tumor-intrinsic manner (i.e., modulating immunogenic cell death), or indirectly by altering the proportions and phenotypes of tumor-associated immune cell subsets[59–61].

In summary, we identified 2 major MIBC subtypes with distinct landscapes across multi-omics levels. The basal subtype was associated with worse survival in the non-NAC patients of the TCGA cohort, but better survival in the NAC patients of the 2 clinical trial cohorts. Each of the 2 major subtypes could be further divided into 2 more subgroups according to the copy number status of chr9 p21.3. Patients with copy number normal of chr9 p21.3 tended to have a higher response rate to PD-L1 blockade therapy and better overall survival. Considering the subtype-specific responses to frontline chemotherapy and immunotherapy, it may be worth exploring the combination therapy in MIBC. Our study has some limitations. First, the sample size of the two clinical trial cohorts was relatively small. More clinical trials with larger sample sizes are necessary to confirm the findings. Second, the history of NAC was known for the patients in the TCGA cohort, but that information was not publicly available for patients in the other 6 retrospective cohorts, which might consist of NAC and non-NAC patients. Third, bulk RNA-seq and microarray GE data are confounded by signals from a mixture of cell populations, making it necessary to collaborate these RNA-based findings with multiplex immunohistochemistry to investigate tumor cell intrinsic changes and its crosstalk with the immune and stromal microenvironments that dictate therapy response. Finally, we are aware that a recent study using the consensus classification method in a different patient cohort revealed basal/squamous tumors exhibited a poor response to chemotherapy[62]. It is unclear if these findings are

related to previous functional studies showing basal cancer stem cell repopulation is a contributing driver to chemoresistance[63]. Future studies will be essential to shed light on the current controversies.

## Methods

**Multi-omics data sets**. The MIBC multi-omics data were generated by TCGA and the level-3 data were downloaded from http://firebrowse.org/. Among the 412 TCGA patient samples, 388 samples with complete somatic mutation, DNA copy number, methylation, and RNA-seq GE data were used for iCluster analysis. Among the 388 samples, 378 samples were collected from patients who received no NAC and 10 samples were collected from patients who received NAC. IMvigor210 RNA-seq GE data ($n = 348$) were from a large phase 2 trial investigating the clinical activity of PD-L1 blockade therapy using atezolizumab in locally advanced and metastatic urothelial carcinoma[6] and made available by Mariathasan et al.[29]. Eight microarray data sets were obtained from ArrayExpress[64] including CIT (Curie Institute, Paris, France) (E-MTAB-1803; n = 85)[14] and from GEO[65] including MDA1 (MD Anderson Cancer Center, Houston, USA) (GSE48277; n = 57)[16], MDA2 (GSE48075; n = 73)[16], CNU (Chungbuk National University, Chungbuk, South Korea) (GSE31507; n = 61)[30], MSK (Memorial Sloan-Kettering Cancer Center, New York, USA) (GSE31684; n = 66)[32], LU (Lund University, Lund, Sweden) (GSE32894; n = 93)[19], MDA MVAC (GSE52219; n = 23)[16] and MDA MVAC + B (GSE69795; n = 38)[31]. In the MDA MVAC study, patients were enrolled for a phase III clinical trial and treated with MVAC (methotrexate, vinblastine, doxorubicin, and cisplatin)[66]. In the MVAC + B study, patients were treated with dose-dense MVAC plus bevacizumab (MVAC + B) in a phase II clinical trial[31]. The other 6 studies were retrospective and the detailed information for chemotherapy was not publicly available. The collection and molecular profiling analysis of the samples were approved by individual Institutional Review Board and informed consent was obtained from each subject as part of previously published studies.

**Integrative clustering and bioinformatics analyses**. iCluster analysis of the MIBC multi-omics data were performed using the iClusterBayes method[28]. To perform the analysis, we processed the MIBC multi-omics data sets to form 4 data matrixes with columns corresponding to the common samples (n = 388) and rows corresponding to omics features. A flowchart summarizing the process was shown in Supplementary Fig. 8. Specifically, the somatic mutation data were summarized by a binary matrix with value 1 (mutation) and 0 (normal) indicating genes' mutation status. A gene was said to be mutated if it contained frameshift deletion/ insertion, in-frame deletion/insertion, missense/nonsense/nonstop mutation, RNA, splice site, or translation start site mutation. The genes with mutation rate ≥2% (3610) were used for iCluster analysis. For the copy number data, we condensed the genomic segments to 6,290 regions using the methods as described by Mo et al.[27]. For the methylation data, we used the beta values that had minimum correlation with corresponding mRNA expression and selected the top 25% (4226) most variable probes for iCluster analysis. For the mRNA expression data, we used the top 25% (5134) most variable genes. In order to fit the model better, we performed logit transformation of the methylation beta values and log2 transformation of the mRNA-seq normalized count values. To find an optimal number of clusters, we tested the cluster number parameter K from 1 to 6. For each K, we ran 36,000 Markov chain Monte Carlo (MCMC) iterations for estimation of model parameters, of which the first 18,000 were discarded as burn-in. After examining the deviance ratio, Bayesian information criterion and heatmap for each possible number of clusters, we found that a 2-cluster solution was optimal (Supplementary Fig. 9) and the samples were divided into 2 clusters according to their latent variable (Z) values (Cluster 1: $Z \geq 0$;Cluster 2: $Z < 0$). Omics features with posterior probability >0.5 were considered as the drivers for sample clustering (Supplementary Fig. 10).

To examine if the GE signature (the 42 basal-luminal genes and the 18 TD genes, respectively) identified by the iCluster analysis had prognostic value, we performed classification analysis using the k-nearest neighbor method in which the TCGA GE data were used as the training data set, and the other GE data were used as the testing data sets. The number of $k$ ($k = 5$ for the 42 genes, and $k = 15$ for the 18 genes) was chosen so that the cross-validation error was minimum in the TCGA GE data. For the microarray GE data, when a gene's expression was measured by multiple probes, we used the one with the largest variance for the classification analysis. To make the data sets comparable for classification analysis, the GE values were standardized across genes for each sample. Pathway and gene set enrichment analysis were performed using GSEA 3.0 (https://www.gsea-msigdb.org/gsea). Gene ontology term enrichment analysis was performed using the DAVID bioinformatics tools (v. 6.7) (https://david.ncifcrf.gov). Cell type enrichment analysis was performed using xCell (2017) (https://xcell.ucsf.edu/).

**Statistics and Reproducibility**. Subtype-specific survival was estimated by Kaplan–Meier method and compared by log-rank test. Fisher's exact test was used to evaluate if somatic mutations were associated with subtypes. Two sample t-test (or Wilcoxon rank-sum test) was used to compare two groups of samples with continuous measurement, and Analysis of Variance (ANOVA) was used for comparison of three or more groups. All the statistical analyses were performed using R 3.6.1 (https://www.r-project.org). P values were two-sided and $P\text{-value} < 0.05$ was considered statistically

significant. If multiple comparisons were involved, p-values were adjusted using Benjamini–Hochberg method.

**Reporting summary**. Further information on research design is available in the Nature Research Reporting Summary linked to this article.

## Data availability

All the data used in this study are publicly available as described in the Methods section. The data behind the figures are available in the supplementary data. The other data will be available from the corresponding author upon request.

## Code availability

The iClusterBayes function in the iClusterPlus package (v. 1.22.0) (https://bioconductor.org/packages/release/bioc/html/iClusterPlus.html) was used for the integrative clustering analysis. R code used for the analyses will be available from the corresponding author upon request.

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

## Acknowledgements
Q.M. is supported in part by the National Cancer Institute Center Core Grants 2P30CA076292 (Cleveland). Q.M. and G.P. are supported in part by National Cancer Institute R01CA181663 (G.P.). Q.M. and K.C. are supported in part by National Cancer Institute R01CA175397 (K.C.).

## Author contributions
Conceptualization, study design, bioinformatics, and biostatistics analyses, Q.M.; Writing original draft, Q.M.; Interpretation of data, manuscript editing, and revision, Q.M., R.L., D.A., G.P., K.C.

## Competing interests
The authors have no competing interests.
