## [Peer Review File · Communications Biology]

Reviewers' comments:

Reviewer #1 (Remarks to the Author):

In recent years, several types of molecular classification of muscle-invasive bladder cancer (MIBC) have been proposed, with the latest one including 5 subtypes based on mRNA expression (Kamoun et al, 2019). Mo et al (JNCI, 2018) proposed yet another classification, designated as tumor differentiation (TD) signature that separated all MIBC tumors only into 2 groups - basal and luminal/ differentiated, based on the expression of 18 genes. Using 5 datasets, that paper has shown that TD was superior to any other classifier (TCGA, MDA, UNC) to predict the overall survival in MIBC.

In the current paper, the authors proposed an extended (integrated) iSignature, now including 42 instead of 18 genes used in the previous analysis (Mo et al, JNCI, 2018). Additionally, this classifier included CNA for the frequently lost chr9p21.3 locus, methylation and somatic mutations. It was concluded that this multi-omics analysis provided better prognostic value for survival and response to frontline therapies - with cisplatin and checkpoint inhibitors.

The results are intriguing, and if this iSignature is easy to generate and is reproducible, this could potentially be very useful, including in clinical applications. However, it is unclear how exactly this new iSignature was derived and was different from the previous TD signature based on 18 genes (Mo et al JNCI, 2018). The TD and iSignature were derived in exactly the same TCGA samples and separated the same clusters (basal and luminal/differentiated) but with 18 or 42 genes, but it is unclear how the additional genes were identified and contributed.

The authors present the same clusters based on all possible omics data. It is unclear how this new classification was improved by the integrative analysis, and what was the contribution of other "omics" to the expression-based TD signature of 18 genes? It would be informative to see a figure showing the structure of this iSignature and the contribution of different omics to it.

How was this iCluster classification derived for non-TCGA samples, that don't have all possible omics data available? Would this signature use the backbone of GE signature (42 genes) and add whatever else is available? Please clarify.

Many mutational analyses are based on targeted panels. Would this be compatible with iCluster approach?

The information about software/code should be provided.

Please provide a supplementary table with TCGA IDs and the assigned iCluster groups.

Reviewer #2 (Remarks to the Author):

The authors analyzed 388 MIBC samples using mutation analysis, DNA copy number, methylation and gene expression analysis. TCGA data was used and iCluster analysis was performed. Two major subtypes are identified (basal/luminal), and a panel of 42 genes were identified to classify independent patient samples (n=844) into basal and luminal subtypes. Basal subtype was correlated to poor survival, but better survival in patient who underwent chemotherapy in two studies. Furthermore, copy number status (loss of chr 9) was correlated to lower response to immunotherapy.

Overall, the analysis presented show high concordance to the TCGA subtypes already identified (86% concordance), and it is questionable if the iCluster analysis and iSubtypes really adds important additional information to the published 5 TCGA subtypes. The authors should

demonstrate that this indeed the case. Furthermore, it is not clear why the authors choose to keep a basal/luminal classification approach – especially when TCGA subtypes and e.g. Consensus MIBC subtypes are published. It is fairly well established that additional granularity is needed to represent the biological subtypes.

Specific points:

1. Page 5 bottom: the authors claim that the iCluster method improve patient classification compared to 5 class TCGA. It is not clear how the authors can make this conclusion based on the current data. It is not enough to simply compare KM plots (and p-values?) – what is the clinical significance of the finding? It is not strange that going from 5 to 2 subgroups will provide better overall statistics.
2. Please clarify how tumors were selected for the study. You mention, “TCGA generated multi-omics data including somatic mutation, GE, DNA copy number and methylation for over 400MIBC samples”. However, you only performed integrative clustering on 388 MIBC samples.
3. There is no reference to Fig. 1B in the text.
4. The authors should look into the associations between clinical characteristics (tstage, histopathological, age, gender, smoking, treatment) and iSubtypes to obtain a better understanding of what the iSubtypes add to already existing knowledge.
5. MIBC iSubtypes should also be compared to the most recent consensus subtypes: Kamoun, A. et al. A Consensus Molecular Classification of Muscle-invasive Bladder Cancer. *Eur. Urol.* (2019) doi:10.1016/j.eururo.2019.09.006.
6. Figure 2: Did you consider if any patients had neoadjuvant treatment that might skew the outcome data? If any patients received NAC you should consider excluding them from the survival analysis.
7. Figure 2D, please clarify in the fig. legend if the p-value ($p=0.0011$) is based on this calculation “luminal-papillary survival curve was significantly separated from the others”.
8. Figure 3 legend. Please correct the order (A)-(B)-(D)-(C) to (A)-(B)-(C)-(D). It seems strange that the fig. 3C-D are described before fig. 3B in the text.
9. Authors place emphasis on the integrative nature of the subtypes, however, when applying the subtyping to imvigor210 data, only the expression of basal and luminal markers are used. The resulting survival difference between basal and luminal indeed also looks similar in 3C and 3D. One might therefore speculate how the proposed subtyping contributes with new information compared to the already published 18-gene signature. Authors should discuss this.
10. Figure 4A-4B: does your findings outperform the already published work by Mariathasan S, et al. “TGFbeta attenuates tumour response to PD-L1 blockade by contributing to exclusion of T cells. *Nature* 554, 544-548 (2018)”? It should be compared directly. Furthermore, was the loss/low expression of this region associated with other characteristics? TMB, TP53 mut, cell cycle activity? If so, could the authors discuss the implications of this.

Response to the reviewers' comments

We would like to thank the reviewers for their insightful and constructive comments. We have taken your comments into account and revised the manuscript accordingly. We updated Figures 2-4 with new data, and added 2 supplementary tables and 6 supplementary figures to the manuscript. Please kindly see our point-to-point responses (in black color) to your comments (in blue color) in the following.

Reviewer #1 (Remarks to the Author):

In recent years, several types of molecular classification of muscle-invasive bladder cancer (MIBC) have been proposed, with the latest one including 5 subtypes based on mRNA expression (Kamoun et al, 2019). Mo et al (JNCI, 2018) proposed yet another classification, designated as tumor differentiation (TD) signature that separated all MIBC tumors only into 2 groups - basal and luminal/ differentiated, based on the expression of 18 genes. Using 5 datasets, that paper has shown that TD was superior to any other classifier (TCGA, MDA, UNC) to predict the overall survival in MIBC.

In the current paper, the authors proposed an extended (integrated) iSignature, now including 42 instead of 18 genes used in the previous analysis (Mo et al, JNCI, 2018). Additionally, this classifier included CNA for the frequently lost chr9p21.3 locus, methylation and somatic mutations. It was concluded that this multi-omics analysis provided better prognostic value for survival and response to frontline therapies - with cisplatin and checkpoint inhibitors.

The results are intriguing, and if this iSignature is easy to generate and is reproducible, this could potentially be very useful, including in clinical applications. However, it is unclear how exactly this new iSignature was derived and was different from the previous TD signature based on 18 genes (Mo et al JNCI, 2018). The TD and iSignature were derived in exactly the same TCGA samples and separated the same clusters (basal and luminal/differentiated) but with 18 or 42 genes, but it is unclear how the additional genes were identified and contributed.

Response: We thank the reviewer for the positive comments. The iSignature was generated through integrative clustering (iCluster) analysis of TCGA somatic mutation, DNA copy number, methylation and gene expression data. By joint statistical modeling of the multi-omics data, the iCluster method clustered cancer samples into major subgroups (subtypes) and at the same time identified genomic features (iSignatures) that contributed to the sample clustering. In the introduction section, we added several sentences to give an untechnical introduction to the iCluster method with comparison to traditional step-wise integrative method (Page 3, 2nd paragraph). In addition, we have provided supplementary Figure S8 to illustrate how the iSignature was derived. The criterion for genomic feature selection was detailed in the Methods section as follows (Pages 17-18 and supplementary Figures S9, S10)

“To find an optimal number of clusters, we tested the cluster number parameter K from 1 to 6. For each K , we ran 36,000 Markov chain Monte Carlo (MCMC) iterations for estimation of model parameters, of which the first 18,000 were discarded as burn-in. After examining the

deviance ratio, Bayesian information criterion⁶⁴ and heatmap for each possible number of clusters, we found that a 2-cluster solution was optimal (Fig. S9) and the samples were divided into 2 clusters according to their latent variable (Z) values (Cluster 1: $Z \geq 0$; Cluster 2: $Z < 0$). Omics features with posterior probability > 0.5 were considered as the drivers for sample clustering (Fig. S10)."

As illustrated in the supplementary Figure S8, we would like to point out that the 42 basal-luminal genes (including the 18 TD genes) were not purposely selected for clustering as what was done by Mo et al. (JNCI, 2018). Instead, these genes were identified by the iCluster algorithm among the other driver genes (2655 genes in total) with similar expression patterns (e.g., low expression in iBasal subtype and high expression in iLuminal subtype or vice versa) (Fig. 1A, Fig. S8). We named the subtypes as iBasal and iLuminal/iDifferentiated because they were characterized by the expression of these 42 basal-luminal genes. Since the 42 basal-luminal genes (or the 18 TD genes) exhibited representative gene expression patterns of the 2655 driver genes and gene expression data were the dominant omics data for studying MIBC molecular subtypes, they were subsequently selected for further classification and prognostic analyses. In contrast, the TD 18 genes were selected because they were involved in cellular differentiation, which was based on biology. To put it simply, the iCluster coincidentally identified the 42 genes including the 18 TD via an unsupervised method, while the 18 TD genes were preselected based on biology knowledge.

The authors present the same clusters based on all possible omics data. It is unclear how this new classification was improved by the integrative analysis, and what was the contribution of other omics to the expression-based TD signature of 18 genes? It would be informative to see a figure showing the structure of this iSignature and the contribution of different omics to it.

Response: We have made a flowchart showing the steps and outcomes for integrative analysis (Supplementary Figure S8). The number of genomic features from each data set contributing to the iCluster is shown on the Figure S8, and the expression pattern of these features are shown in Figure 1A. Compared to the TD KM survival curves, the iSubtype KM curves are more widely separated and the p-value is more significant (Fig. 2B, 2C), demonstrating that the integrative analysis achieves a better result than the TD gene expression-based analysis. The iSubtypes were primarily driven by mRNA expression (2655 features) and methylation (2983 features), compared to the 114 copy number features and 18 mutated genes (Figure S8). Since mRNA expression and methylation patterns were highly correlated, we thought it was possible to select some representative genes for classification and prognostic study. We found that the 42 classical basal-luminal genes including the 18 TD genes were among the 2655 driver genes. Therefore, we used them for the following classification and prognostic analyses. In addition, as demonstrated in the analysis of IMvigor210 data, the copy number change on chr9 p21.3 were also useful to further classify patients into subgroups responding differently to PD-L1 blockade therapy.

How was this iCluster classification derived for non-TCGA samples, that don't have all possible omics data available? Would this signature use the backbone of GE signature (42 genes)

and add whatever else is available? Please clarify.

Response: For the non-TCGA samples, the k-nearest neighbor method was used for sample classification. Since most of the MIBC subtypes were based on gene expression signature and the gene expression data were the dominant genomic data type, we used the 42 genes as well as the 18 TD genes as the backbone for classification. We clarify that in the Methods section as follows.

“To examine if the GE signatures (the 42 basal-luminal genes and the 18 TD genes, respectively) identified by the iCluster analysis had prognostic value, we performed classification analysis using the k-nearest neighbor method in which the TCGA GE data were used as the training data set, and the other GE data were used as the testing data sets. The number of k (k=5 for the 42 genes, and k=15 for the 18 genes) was chosen so that the cross-validation error was minimum in the TCGA GE data.”

Many mutational analyses are based on targeted panels. Would this be compatible with iCluster approach?

Response: In terms of approaches, targeted panels based mutational analyses are not compatible to the iCluster approach. In the targeted panels based mutational analyses, known genes are selected for mutation analysis. The researchers may evaluate if targeted genes' mutations are associated with specific subtypes. In contrast, the iCluster method integrates multi-omics including somatic mutation data via joint statistical modeling. The genes used for iCluster analysis are unbiasedly selected and the iCluster algorithm identifies genes that significantly contribute to sample clustering. Interestingly, important genes (e.g., TP53, KDM6A, RB1, FGFR3, PIK3C2A) that are usually included in the targeted panels are identified as the driver genes in our iCluster analysis.

The information about software/code should be provided.

Response: The information about software/code used for this study is now described in the revised manuscript (page 20).

Please provide a supplementary table with TCGA IDs and the assigned iCluster groups.

Response: As suggested by the reviewer and required by *Communications Biology*, all the data behind the figures including subtype information for the TCGA, IMvigor210 and the other cohorts are provided in the supplementary data 1 to 7.

Reviewer #2 (Remarks to the Author):

The authors analyzed 388 MIBC samples using mutation analysis, DNA copy number, methylation and gene expression analysis. TCGA data was used and iCluster analysis was performed.

Two major subtypes are identified (basal/luminal), and a panel of 42 genes were identified to classify independent patient samples (n=844) into basal and luminal subtypes. Basal subtype was correlated to poor survival, but better survival in patient who underwent chemotherapy in two studies. Furthermore, copy number status (loss of chr 9) was correlated to lower response to immunotherapy.

Overall, the analysis presented show high concordance to the TCGA subtypes already identified (86% concordance), and it is questionable if the iCluster analysis and iSubtypes really adds important additional information to the published 5 TCGA subtypes. The authors should demonstrate that this indeed the case. Furthermore, it is not clear why the authors choose to keep a basal/luminal classification approach – especially when TCGA subtypes and e.g. Consensus MIBC subtypes are published. It is fairly well established that additional granularity is needed to represent the biological subtypes.

Response: We thank the reviewer for these insightful comments. The major reason that the MIBC samples were clustered to two major subtypes was because the multi-omics data fitted the iCluster model the best when the samples were divided into two clusters. In the revised manuscript, we provide supplementary Figure S9 to illustrate why the 2-cluster solution was chosen. We tested 2- to 7-clusters solutions (corresponding to cluster parameter $K = 1$ to 6, respectively) for the multi-omics data. From Figure 9S, we can see the deviance ratio is the maximum and the Bayesian information criterion is the minimum when the cluster parameter K is 1, indicating that the data fitted the iCluster model best and a 2-clusters solution was the best.

As described in the manuscript (the first paragraph on page 3), both the TCGA and consensus subtypes were actually made of two major subtypes basal and luminal. For both TCGA and consensus subtypes, in term of overall survival, only the luminal papillary subtype was significantly different from the others, and there was not statistically different among the other subtypes (Fig. 2D, 2E). Therefore, from a clinical point of view, a 2-subtype solution may be sufficient.

Additionally, using the copy number information of chr9 p21.3 (or expression of the 3 genes CDKN2A/2B and MTAP (G3)), we further divided the subtypes into 9p21.3 normal (9p21.3N/G3High) or loss group (9p21.3L/G3Low). We showed these sub-groups had significantly different response rates to PD-L1 blockade therapy and overall survival. To our best knowledge, these discoveries are new, which certainly add values to our understanding of bladder cancer. In addition, to make it useful, we derived a 42 gene panel for classification of bladder cancer samples, which will be much easier to use in practice than those classifiers involved in thousands of genes and complicated algorithms. Therefore, we firmly believe that this manuscript provides important findings for our understanding of bladder cancer, which will be interesting to research community.

Specific points:

1. Page 5 bottom: the authors claim that the iCluster method improve patient classification compared to 5 class TCGA. It is not clear how the authors can make this conclusion based on the current data. It is not enough to simply compare KM plots (and p-values?) â “ what is the clinical significance of the finding? It is not strange that going from 5 to 2 subgroups will provide better overall statistics.

Response: On page 5 (now page 6 of the revised manuscript), we actually compare the iSubtypes with the tumor differentiation (TD) gene expression subtypes. Since both iSubtypes and TD subtypes have two sub-groups, they are comparable. We agree with the reviewer that it is not enough to simply compare KM plots. Therefore, we compared the differences of median survival of the two subtypes and the corresponding p-values. As suggested by the reviewer, we removed 10 patients with NAC from the survival analysis, and updated paragraph as the following.

“In the chemotherapy-naïve patients, the iBasal subtype was significantly associated with a worse overall survival, compared to the iLuminal subtype (Fig. 2B, $P = 0.00042$). Their survival curves (median survival, iBasal: 22.5 vs. iLuminal: 54.9 months; Fig. 2B) were more widely separated than the curves of the tumor differentiation subtypes¹⁷ (median survival, basal: 25.6 vs. differentiated: 41.7 months; Fig. 2C, $P = 0.015$), demonstrating an improvement of patient classification in terms of survival.”

Because the iSubtype KM curves are more widely separated than the TD subtype KM curves, we conclude that the integrative clustering analysis improve patient classification.

Since TCGA subtypes have 5 sub-groups, we agree with the reviewer that it might not be a fair comparison between a 2-class and 5-class systems in terms of survival. Instead, we pointed out in the result section that only the luminal-papillary survival curve was significantly separated from the other survival curves and there was not statistically different in overall survival among the Basal-squamous, Luminal, Luminal-infiltrated and Neuronal subtypes (page 6, the bottom).

“Although there were 5 TCGA subtypes, only the luminal-papillary survival curve was significantly separated from the others (Fig. 2D, overall $P = 0.00047$). There was not statistically different in overall survival among the basal-squamous, luminal, luminal-infiltrated and neuronal subtypes (Fig. 2D, $P = 0.3$).”

2. Please clarify how tumors were selected for the study. You mention, â œTCGA generated multi-omics data including somatic mutation, GE, DNA copy number and methylation for over 400MIBC samplesâ . However, you only performed integrative clustering on 388 MIBC samples.

Response: There were 412 MIBC samples. However, only 388 samples had complete somatic mutation, GE, DNA copy number and methylation data. The iCluster method requires complete data from all the platforms and does not allow samples with missing data. Therefore, only 388 samples were used for iCluster analysis. We have now included in the methods section (page 13) explaining why 388 samples were used for iCluster analysis as follows.

“The MIBC multi-omics data were generated by TCGA and the level-3 data were downloaded from <http://firebrowse.org/>. Among the 412 TCGA samples, 388 samples with complete somatic mutation, DNA copy number, methylation and RNA-seq GE data were used for iCluster analysis.”

3. There is no reference to Fig. 1B in the text.

Response: Thanks for catching this mistake. The Fig. 1A should be Fig. 1B on the bottom of page 5. We have fixed the mistake as follows (page 5, the first paragraph).

“Gene ontology (GO) term enrichment analysis showed that methylation cluster M2 was most enriched with genes involved in cell adhesion/motion/morphogenesis, response to wounding/organic substance, leukocyte activation, and skeletal/urogenital system development, while no significant biological process was found in cluster M1 (Summarized in Fig. 1B and Supplementary data 1).”

4. The authors should look into the associations between clinical characteristics (tstage, histopathological, age, gender, smoking, treatment) and iSubtypes to obtain a better understanding of what the iSubtypes add to already existing knowledge.

Response: In the revised manuscript, we provide Supplementary Table S1 to summarize the patient characteristics for the iSubtypes. In addition, we performed multivariate Cox regression analysis including baseline characteristics: pathological stage (or T stage), age, gender and smoking status into the models. Since pathological stages and T stages were highly correlated, they were included in separate multivariate models. The iSubtype is still statistically significant ($P = 0.0086$ if T stage is included in the multivariate model; or $P = 0.003$ if Pathological stage is included in the multivariate model). These results show that iSubtype is an independent predictor of overall survival. These results are summarized as follows on page 6.

“The iSubtypes were significantly associated with pathologic stage ($P = 0.011$), T stage ($P = 0.0017$) and gender ($P = 0.027$) (Table S1). However, multivariate Cox regression analysis showed that the iSubtype was an independent predictor of overall survival when the baseline variables including age, gender and smoking status, pathological stage (or T stage) were included into the models (Table S2).”

5. MIBC iSubtypes should also be compared to the most recent consensus subtypes: Kamoun, A. et al. A Consensus Molecular Classification of Muscle-invasive Bladder Cancer. *Eur. Urol.* (2019) doi:10.1016/j.eururo.2019.09.006.

Response: We have compared the iSubtypes to the consensus subtypes. The results are shown in the updated Figure 2A and Figure 2E and described on pages 6-7.

6. Figure 2: Did you consider if any patients had neoadjuvant treatment that might skew the outcome data? If any patients received NAC you should consider excluding them from the survival analysis.

Response: There were 10 patients receiving NAC among the 388 cases. Six patients were clustered to the iBasal subtype and 4 patients were clustered to the iLuminal subtypes. To avoid the effect of NAC, we re-analyzed the data excluding the 10 NAC patients and updated the results (Fig. 2 and Supplementary Fig. S2). We can see that the results of NAC patients are quite similar whether the 10 NAC patients are excluded or not, probably due to their small number.

7. Figure 2D, please clarify in the fig. legend if the p-value ($p=0.0011$) is based on this calculation "luminal-papillary survival curve was significantly separated from the others".

Response: The p-value on Fig. 2D is the overall p-value, which has been clarified in the updated Fig. 2D. In response to the reviewer, we have also added the p-value for Luminal-papillary vs. the others to the Fig. 2 legend.

8. Figure 3 legend. Please correct the order (A)-(B)-(D)-(C) to (A)-(B)-(C)-(D). It seems strange that the fig. 3C-D are described before fig. 3B in the text.

Response: The Fig. 3 has been updated and described in order in the text. We moved 2 figures from the original Fig. 4 to the updated Fig. 3. Now the updated Fig. 3 has 6 sub figures.

9. Authors place emphasis on the integrative nature of the subtypes, however, when applying the subtyping to imvigor210 data, only the expression of basal and luminal markers are used. The resulting survival difference between basal and luminal indeed also looks similar in 3C and 3D. One might therefore speculate how the proposed subtyping contributes with new information compared to the already published 18-gene signature. Authors should discuss this.

Response: Gene expression data were the most dominant genomics data used to define the MIBC subtypes. The reason that we were only able to use the expression of basal and luminal markers to classify the IMvigor210 samples was because no other data types were available, other than gene expression. In fact, we aimed to identify signature that can be easily applicable to other patient cohorts for classification analysis, which primarily contain gene expression data. After the integrative analysis, we found that the gene expression and methylation were the major driving force for sample clustering and they were highly correlated (Figure 1A). Therefore, we thought it was possible to derive expression signatures that can be applied to other cohorts. By carefully examining the driver genes, we found that the classical basal, luminal and the tumor differentiation genes were all identified as the driver genes. Therefore, we applied the basal-luminal and tumor differentiation signatures to IMvigor210 and other gene expression data sets to examine their prognostic values.

It is really interesting that the unsupervised integrative clustering resulted in two subtypes that are highly concordant to a biologically-supervised clustering approach (TD). The Fig. 3B and

3C shows the survival curves for the subtypes defined by the basal-Luminal (42) genes, and the TD (18 genes), respectively. We agree that the survival curves look similar. The purpose of showing the results is to demonstrate the prognostic power of the two signatures, instead of demonstrating that one signature is superior to the other. We have discussed this in the revised manuscript (the 2nd paragraph on page 14 and the 1st paragraph on page 15).

10. Figure 4A-4B: does your findings outperform the already published work by Mariathasan S, et al. *α*TGFβ attenuates tumour response to PD-L1 blockade by contributing to exclusion of T cells. *Nature* 554, 544-548 (2018)? It should be compared directly. Furthermore, was the loss/low expression of this region associated with other characteristics? TMB, TP53 mut, cell cycle activity? If so, could the authors discuss the implications of this.

Response: We would like to thank the reviewer for this great suggestion. We have compared our results with the published work by Mariathasan et al. in terms of subtype-specific response rates to PD-L1 blockade therapy, PD-L1 expression level and survival and showed these results on the updated Figure 4. Mariathasan et al. reported that 3 immune cell subtypes, 3 tumor-immune phenotypes, and the Lund subtypes were associated with responses to PD-L1 blockade therapy. From Figure 4, we can see that our results are as good as the other subtyping methods. More importantly, our classification method is much simpler, which should be much easier to be used for stratification of MIBC patients into clinically meaningful groups for target therapy. These results are summarized on pages 8 and 9 of the updated manuscript.

The loss/low expression of chr9 p21.3 was not associated TMB (See supplementary Fig. S1), but associated with TP53 mutation and cell cycle activity. We have added these results to the manuscript (see the bottom on page 4, the top on page 5 and page 8) and discussed them in the discussion section (pages 13, 14).

REVIEWERS' COMMENTS:

Reviewer #1 (Remarks to the Author):

The authors provided satisfying responses and a relevant update of their paper.

Ludmila Prokunina-Olsson

Reviewer #2 (Remarks to the Author):

The authors have updated the manuscript and included new analysis as requested in my first review.

One question remaining is the authors use of "chemotherapy naïve" patients from the TCGA cohort (results and discussion). This may be misleading, as most patients in the TCGA cohort have received chemotherapy during the disease course I guess - otherwise this cohort is highly biased. How should the reader interpret this? If the authors mean that no NAC has been used then this should be stated. This may make the tumors chemotherapy-naïve if taken at TURB before NAC, but I guess most tumors are obtained before treatment anyway.

In principle the biological mechanisms underlying response to chemotherapy should be similar in early- (NAC) and late (metastatic) treatment setting.

Response to the reviewers' comments

We would like to thank the reviewers for their constructive comments, which have led to improvement of our manuscript. Please kindly see our point-to-point responses (in black color) to your comments (in blue color) in the following.

REVIEWERS' COMMENTS:

Reviewer #1 (Remarks to the Author):

The authors provided satisfying responses and a relevant update of their paper.

Ludmila Prokunina-Olsson

Reviewer #2 (Remarks to the Author):

The authors have updated the manuscript and included new analysis as requested in my first review.

One question remaining is the authors use of “chemotherapy naïve” patients from the TCGA cohort (results and discussion). This may be misleading, as most patients in the TCGA cohort have received chemotherapy during the disease course I guess - otherwise this cohort is highly biased. How should the reader interpret this? If the authors mean that no NAC has been used then this should be stated. This may make the tumors chemotherapy-naïve if taken at TURB before NAC, but I guess most tumors are obtained before treatment anyway.

In principle the biological mechanisms underlying response to chemotherapy should be similar in early- (NAC) and late (metastatic) treatment setting.

Response: We would like to thank the reviewer for the insightful comments. The chemotherapy naïve patients from the TCGA cohort refer to the patients who received no neoadjuvant chemotherapy (NAC), in contrast to the patients who received NAC in the two clinical trials. To avoid confusion, we have replaced “chemotherapy naïve” with “no NAC” in the revised manuscript. For example, in the abstract, the revised sentence is

“The basal subtype was associated with worse overall survival in patients receiving no neoadjuvant chemotherapy (NAC), but better overall survival in patients receiving NAC in two clinical trials.”.

In result section (page 10, the last paragraph), we revised the related sentence as

“Two microarray data sets were from clinical trials that were designed to evaluate the effect of NAC (MVAC or MVAC+B) on clinical outcomes and the impact of MIBC subtypes”.